# Non-reciprocal and non-Newtonian mechanical metamaterials

Lianchao Wang [1,2], Julio A. Iglesias Martínez[2], Gwenn Ulliac[2], Bing Wang [1] ✉, Vincent Laude [2] & Muamer Kadic[2] ✉

Non-Newtonian liquids are characterized by stress and velocity-dependent dynamical response. In elasticity, and in particular, in the field of phononics, reciprocity in the equations acts against obtaining a directional response for passive media. Active stimuli-responsive materials have been conceived to overcome it. Significantly, Milton and Willis have shown theoretically in 2007 that quasi-rigid bodies containing masses at resonance can display a very rich dynamical behavior, hence opening a route toward the design of non-reciprocal and non-Newtonian metamaterials. In this paper, we design a solid structure that displays unidirectional shock resistance, thus going beyond Newton's second law in analogy to non-Newtonian fluids. We design the mechanical metamaterial with finite element analysis and fabricate it using three-dimensional printing at the centimetric scale (with fused deposition modeling) and at the micrometric scale (with two-photon lithography). The non-Newtonian elastic response is measured via dynamical velocity-dependent experiments. Reversing the direction of the impact, we further highlight the intrinsic non-reciprocal response.

In physics, a non-Newtonian fluid is characterized by a stress- and velocity-dependent viscosity, or more generally by a mechanical response[1]. In such a fluid, viscosity can drastically change under a dynamically applied force, effectively turning the body response into either more liquid or more solid, and leading to a drastic stiffening feeling for an external excitation[2]. From the physical or mechanical point of view, the viscosity (the deformation by shear forces) of non-Newtonian fluids depends on the shear rate[3]. Generally, in a Newtonian fluid, the relation between shear stress and shear rate is linear[4–7]. In a non-Newtonian fluid, however, this relationship can in principle assume any shape, from exponential to logarithm-like[1]. The fluid thus exhibits a time-dependent viscosity response[8,9]. Although non-Newtonian fluids have a long history, few related applications have been reported[1]. Furthermore, non-Newtonian fluids are acutely sensitive to temperature[10].

Metamaterials with tunable mechanical properties[11–14], in particular having a rich dynamical behavior[15–22] or exhibiting non-reciprocity[23,24], are attractive for numerous applications, such as

control of the propagation of acoustic and elastic waves[25–43]. Beyond the homogenization limit, the idea of dynamic shear, bulk, or mass density is commonly used and accepted for waves[44]. This dependence is often related to internal resonances and is not easily distinguished from the homogenized descriptions. The pioneering work by Liu et al.[25] showed that a negative modulus can be observed at resonance. Massive models resulting in dynamical behaviors were then proposed, e.g., to obtain negative mass density[45–48], negative Young's modulus[26,48], or negative bulk modulus[49,50]. Based on these effective models, many metamaterials with excellent acoustic characteristics have been designed and studied[51–54]. Moreover, both Maxwell and Kelvin–Voigt materials show additional resistance to fast deformations[15–17], providing models that can be extended to metamaterials[18–22]. For instance, based on the Kelvin–Voigt model, Janbaz et al. used bi-beams composed of a hyperelastic beam and a visco-hyperelastic beam in parallel as the elements to design mechanical metamaterials with varying deformation characteristics and strain rate-dependent mechanical response[55]. In addition,

[1]National Key Laboratory of Science and Technology on Advanced Composites in Special Environments, Harbin Institute of Technology, 150001 Harbin, P.R. China. [2]Université Franche-Comté, CNRS, Institut FEMTO-ST, Besançon 25000, France. ✉e-mail: wangbing86@hit.edu.cn; muamer.kadic@univ-fcomte.fr

autonomous controllers were used to design topological mechanical metamaterials which are not constrained by Newtonian dynamics[56]. The interaction between mechanical instabilities and viscoelasticity was discussed and utilized by Dykstra et al.[57] to obtain loading rate-dependent functionalities. Significantly, Milton and Willis[58] have proposed in the context of homogenized media an extension of Newton's second law. Their model proposes a new description for rigid bodies with hidden inclusions.

In fluids and solids, the principle of reciprocity in acoustics and elastodynamics codifies a relation of symmetry between action and reaction, which is hard to overcome when time-reversal symmetry applies[24]. Materials exhibiting nonreciprocity, also termed Willis coupling for elastic waves in solids[59–61], however, promise to regulate waves unidirectionally[62–74]. Coulais et al. showed that it is possible to break reciprocity in slightly non-linear static systems, obtaining mechanical metamaterials with asymmetric output under excitation from different directions[23]. Trainiti and Ruzzene[75] broke mechanical reciprocity by dynamically modulating the Young's modulus and the density of the material. The effect on acoustic nonreciprocity of spatio-temporally varying material properties was also investigated[66,76]. Moreover, the non-reciprocal transmission of acoustic waves can be realized by dynamic boundary conditions[77] and in an air flow[65]. Li et al. studied the effect of initial stress on the propagation of elastic waves in nonlinear metamaterials and proved the occurrence of non-reciprocal transmission[78]. Fang et al. introduced a nonlinear local resonance unit and demonstrated acoustic non-reciprocity at low frequencies and for small sizes[79]. In fact, using a single raw material to design metamaterials exhibiting non-reciprocity and velocity-dependent mechanical response is still a challenge.

In this work, we introduce the concept of stretchable and compressible elastic metamaterials that mimic the response of non-Newtonian fluids and are at the same time non-reciprocal. Significantly, they can be tuned at will to obtain the desired effective stiffness. Similarly to Milton and Willis[58], though for an elastic and not a rigid body, we show that the dynamical response of the metamaterials does not follow the effective description of standard Cauchy–Newton elastic bodies. First, a general effective mass–spring model for non-reciprocal and non-Newtonian elasticity is proposed. Then, a specific example is used to validate the effectiveness of this model for the design of 2D and 3D metamaterials. 3D printed samples are used to obtain experimentally the non-reciprocal and non-Newtonian mechanical response and to compare it to numerical simulations performed with finite element analysis. Finally, we discuss possible applications, e.g. to satellite docking or transportation.

## Results

### Effective model of non-reciprocal and non-Newtonian metamaterials

A simple mass–spring toy model is first introduced for illustration in Fig. 1a. Under resonant external stimuli, the spring–mass system oscillates around its equilibrium position and generates a periodic restoring force. Obviously, the amplitude of the restoring force is highly dependent on both the frequencies of the external stimuli and on the inherent properties of the mass–spring system (i.e., the mass and the spring constants). Herein, based on the principle of local resonance[25], we propose the effective mass–spring model for non-reciprocal and non-Newtonian metamaterials depicted in Fig. 1b. This effective model includes four parts: the main system, converters, subsystems, and clipping boundaries. The main system consists of a large mass $M$ and of a spring $K$. The converters provide a frequency-dependent transformation of the direction of motion between the main system and the subsystems. In fact, the design of the converters is an essential and flexible task influencing the whole effective model due to the primary and exclusive connection that exists between the main effective mass $M$ and the subsystems. Importantly, the subsystems are

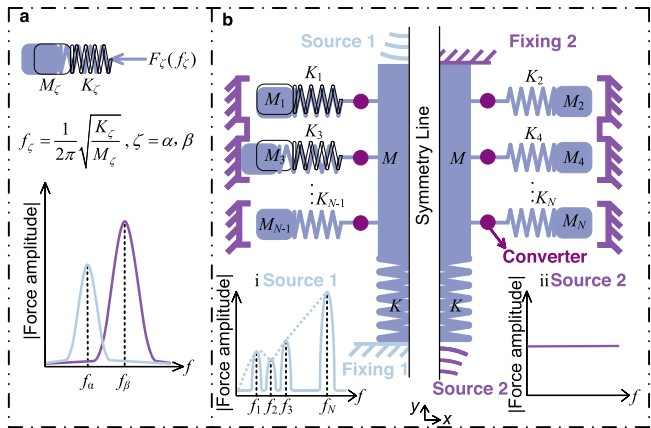

**Fig. 1 | Toy model of the metamaterial. a** For a simple mass-spring model, the response function is frequency dependent under time-harmonic excitation. Resonance is observed at $f_\zeta = \sqrt{K_\zeta / M_\zeta}/2\pi, \zeta = \alpha, \beta$. The amplitude of the restoring force $F_\zeta$ is highly dependent on both the frequencies of the external stimuli and on the inherent properties (mass $M_\zeta$ and stiffness $K_\zeta$) of the mass–spring system. **b** The effective mass–spring model of the non-reciprocal and non-Newtonian metamaterial includes four parts: the main system is composed of a large mass $M$ and a spring $K$, of converters, of $N$ resonant subsystems ($M_i, K_i$), $i = 1 \cdots N$, and of $N$ clipping boundaries. The function of the converters is to provide a frequency-dependent transformation of the direction of motion between the main system and the subsystems. The mechanical properties in this model depend on both the direction and on the frequency spectrum of the external excitation. For example, when the source is located at the top (source 1) and the bottom is clamped, the amplitude of the restoring force is highly dependent on frequency $f$, thanks to the influence of the resonances of the subsystems (Fig. 1b-i). In contrast, when the source is located at the bottom (source 2) and the top is clamped, the rigid mass $M$ is motionless and resonances are not excited. In this case, no frequency-dependent behavior is expected (Fig. 1b-ii).

not directly connected to the spring $K$ but are only connected to the mass $M$, implying that the spring is not able to induce any motion or vibration of the subsystems. The details of the converter will be discussed in the next section. The subsystems are composed of $N$ pairs of masses $M_1, M_2 \ldots M_N$ and springs $K_1, K_2 \ldots K_N$.

Herein two situations are taken into consideration. The first situation, termed source 1, is that the external stimuli come from the top of the structure whereas the bottom remains fixed (see left panel of Fig. 1b). On the one hand if the source does not contain resonant frequencies (i.e. if the frequency spectrum of the source does not overlap with the resonant frequencies of the subsystems), the force amplitude of the whole system depends on the spring $K$ and the subsystems move as a single rigid body along the source direction, as depicted for the deformed subsystem ($M_1, K_1$). On the other hand, if the spectrum of the external source contains resonant frequencies of the subsystems, it induces the resonance of one or more of them. In the latter case, the resonant subsystems undergo rigid motion along the direction of the source and simultaneously elastic vibrations along another direction, as depicted for the deformed subsystem ($M_3, K_3$). The details of the transformation of the direction between the external source and the subsystems will be discussed in the following section. The elastic vibrations of the subsystem can cause it to touch the fixed boundary, which significantly influences the mechanical response of the entire system. In theory, the amplitude of the reaction force of the whole system changes continuously as a function of the spectrum of the source, exciting the $N$ resonant frequencies ($f_1, f_2, f_3 \ldots f_N$) depicted in Fig. 1b-i. It is clear that the amplitude of the reactions at the resonant frequencies is significantly larger than off-resonance and is the origin of the non-Newtonian behavior. Moreover, it is worth emphasizing that the enhancement of the force amplitude at the resonant frequencies depends on two decisive factors, namely the resonance of the

subsystems and the influence of the fixed boundaries, which will be further discussed later with a specific example.

The second situation that is considered, termed source 2, is that the external stimuli come from the bottom of the structure whereas the top remains fixed (see right panel of Fig. 1b). In this case, if we assume that the main beam $M$ is an ideally rigid body, then regardless of the frequency spectrum of the source, the mechanical response of the entire system is constant, since source 2 is not able to induce the resonance of the subsystems, as depicted in Fig. 1b-ii. Importantly, the resonance of the spring $K$ is ignored under this setting. As a whole, the mechanical response of the structure acutely depends on both the frequency spectrum and the direction of application of the external stimuli, which is the essence of the present model of non-reciprocal and non-Newtonian mechanical metamaterials.

## Equations of motion for the toy model system

The toy model can be formulated as follows. Source 1 is characterized by an applied force on the main mass that is a function of time, $F_1(t)$. For instance, that applied force can be time-harmonic, i.e. of the form $F_1(t) = A \exp(\imath \omega t)$ with driving angular frequency $\omega$ and amplitude $A$. The center of mass of the main system considered a rigid body, has displacements $(x, y)$. By symmetry, we assume $x = 0$ at all times. The displacements of the sub-systems are represented by degrees of freedom $(x_i, y_i)$, $i = 1 \cdots N$. Let us first consider the case that no internal resonances of the sub-systems are excited, i.e., $\omega$ remains far from any of the resonance frequencies $\omega_i = \sqrt{K_i / M_i} = 2\pi f_i$. Then we can assume $x_i = 0$ and $y_i = y$. The motion of the main mass satisfies the dynamical equation

$$\left( M + \sum_{i=1}^{N} M_i \right) \ddot{y} + Ky = F_1(t), \tag{1}$$

Damping is ignored in the whole system and the masses of all springs are neglected for simplicity. Equation (1) can easily be solved as a function of time.

Next, we include the motion of the sub-systems, i.e. frequency $\omega$ can approach one or several of the resonance frequencies $\omega_i$. Each of the sub-systems satisfies a dynamical equation

$$M_i \ddot{x}_i + K_i x_i = F_i^r(t) = \gamma_i(\omega) A \exp(\imath \omega t). \tag{2}$$

$F_i^r(t)$ is the $x$-component of the reaction force introduced by the resonance of the subsystem. $\gamma_i(\omega)$ is a conversion factor of source 1 to the motion of sub-system $i$, expressing the assumed linearity of the reaction force with the applied force. As a note $\gamma_i(\omega)$ may not be precisely known if the properties of the converter are not defined. The reaction forces act on the main rigid mass, but we also have to include in the model the forces resulting from contact of the sub-systems with the fixed external boundaries, noted $F_i^b(t)$. The $F_i^b(t)$ are again functions of the applied force, e.g. depend on $A$ and $\omega$, but this time in a non-linear way. The motion of the main mass now satisfies

$$\left( M + \sum_{i=1}^{N} M_i \right) \ddot{y} + Ky = F_1(t) + \sum_{i=1}^{N} \alpha_i F_i^r(t) + \sum_{i=1}^{N} F_i^b(t), \tag{3}$$

Factors $\alpha_i$ are introduced to express the linear relationship of the $y$-component of the reaction forces with their $x$-component.

All in all, Eqs. (2) and (3) provide $N + 1$ equations for the $N + 1$ unknowns $(y, x_i)$ and the model can be solved. In practice, however, it would be necessary to provide accurate values for parameters $\gamma_i(\omega)$ and $\alpha_i$, and an expression for forces $F_i^b(t)$. Hence, the toy model is used here only to discuss the origin of the dynamical non-Newtonian properties, but we use numerical simulations based on continuum linear mechanics in the following.

Source 2 can also be characterized by an applied force, but on the main spring and not on the main mass, which is a function of time, $F_2(t)$. Denoting $\xi$ the elongation of the main spring, Hooke's law yields

$$K\xi = F_2(t). \tag{4}$$

Applied force $F_2(t)$ is transferred to the main mass but the latter does not move, because of the assumption of rigidity and of the application of a clamping boundary condition at the top, hence $y = 0$ and $y_i = 0$. This is of course a strong assumption requiring a very soft main spring and a very rigid main mass, but a direct expression of the mechanical non-reciprocity of the system.

## Design philosophy of converters and metamaterial

As discussed in the previous section, the converters are essential parts of the proposed non-reciprocal and non-Newtonian metamaterial model. In general, the design of a converter is an open question since we simply have to provide a frequency-dependent transformation of the direction of motion between the main system and the subsystems. In mechanical design, it would be easy to realize the required function by employing multi-body dynamical systems, such as gear and rack systems. However, there are still many challenges to overcome in the design of periodic porous metamaterials with embedded multi-body systems. For instance, the distance between each flexible component is tough to control precisely, especially at the micron scale[80]. Therefore, a more straightforward method to design the converters is required.

In the field of kinematics, an external force induces the motion of an object, generating rigid-body translation and/or rotation. The nature of the induced motion is dominated by the relative position of the external force direction and the centroid of the object. For example, the object will merely translate along the direction of the external force in case the force passes through the objective centroid. If not, the object will translate and rotate simultaneously, as depicted in Fig. 2a. It is worth noting that the rotation in the latter circumstance can cause additional displacement in a direction different from the direction of the external force, which is the core requirement for the converter.

Therefore, we designed the converter considering the off-axis loading of a tilted cantilever with a concentrated mass at the free end, as shown in Fig. 2b. The angle between the external force $\mathbf{F}$ and the tilted cantilever is noted $\delta$. The force $\mathbf{F}$ can be decomposed into two components, $F_1 = F \sin \delta$ and $F_2 = F \cos \delta$. In fact, $F_1$ and $F_2$ are, respectively, able to cause the transverse and the longitudinal vibration of the tilted cantilever. Nevertheless, the longitudinal resonant frequencies induced by $F_2$ are larger than the transverse resonant frequencies induced by $F_1$, so longitudinal vibrations will be ignored in this work. Regarding transverse vibrations, the velocity $\mathbf{v}$ of the free end of the tilted cantilever is decomposed into two components, $v_1 = v \cos \delta$ (perpendicular to $\mathbf{F}$) and $v_2 = v \sin \delta$ (parallel to $\mathbf{F}$). The velocity component $v_1$ induces a displacement that is perpendicular to the original external force $\mathbf{F}$, under the circumstance that the frequency of the external force matches the resonance frequency of the tilted cantilever, as illustrated at the bottom of Fig. 2b. In brief, the off-axis loading tilted cantilever is able to offer a frequency-dependent conversion of the direction of motion of the external source toward a different direction, which is the basic and important function required of the converter. In addition, the resonant frequencies of the cantilever with concentrated mass at the free end can be calculated by the beam equation $f_0 = \frac{1}{2\pi} \sqrt{\frac{Ewt^3}{4L^3(M + \rho t w L)}}$, where $f_0, M, \rho, t, w, L, E$ are the resonant frequency, the concentrated mass, the density of mass, the thickness, the width, the length and the elastic modulus of the cantilever, respectively.

We then propose the 2D unit cell of a non-reciprocal and non-Newtonian metamaterial depicted in Fig. 2c. The unit cell consists of a

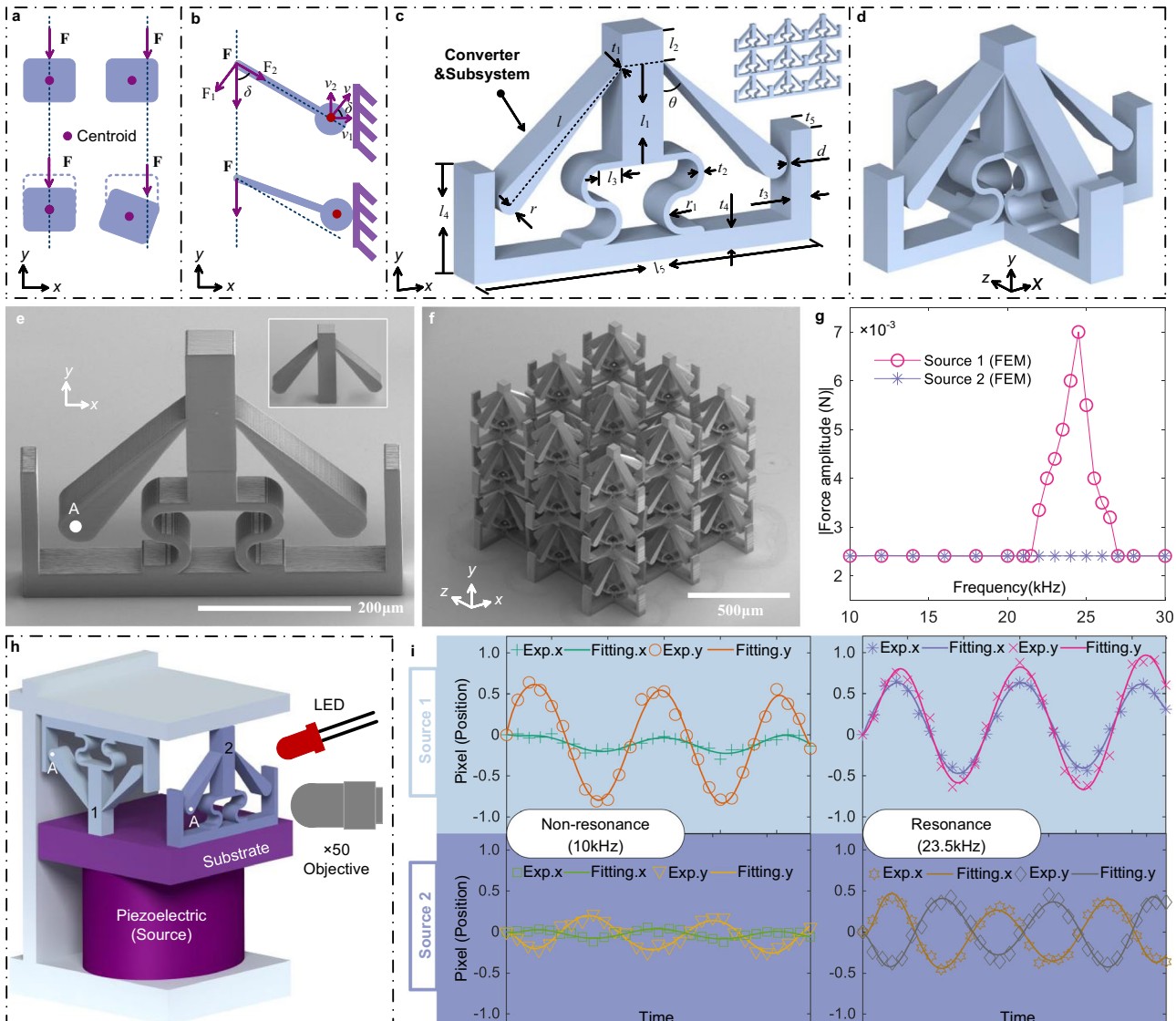

**Fig. 2 | Design model for converters and metamaterials. a** The simple principle of rotation and torque is induced by a unidirectional force **F** applied on a non-centrosymmetric object. **b** The design of the converter uses off-axis loading of tilted cantilevers with a concentrated mass at the end. The arms are deflected and clipping of the mass is expected on the side wall. The response is exalted at the resonance frequency of the ball-beam system (mass-spring). $v$: the velocity of beam lateral vibration. **c** A planar example (2D) of a non-reciprocal and non-Newtonian unit cell is shown. **d** An example of a 3D unit cell inspired by the previous 2D unit cell is depicted. **e**, **f** Micro-scale 2D planar unit cell and 3D metamaterial fabricated by two-photon lithography 3D direct laser writing. The geometry parameters for 2D unit cell and 3D metamaterial are $t_1 = 4\,\mu m$, $t_2 = 10\,\mu m$, $t_3 = 15\,\mu m$, $t_4 = 30\,\mu m$, $t_5 = 50\,\mu m$, $r = 20\,\mu m$, $d = 20\,\mu m$, $l = 130\,\mu m$, $l_1 = 126\,\mu m$, $l_2 = 70\,\mu m$, $l_3 = 25\,\mu m$, $l_4 = 130\,\mu m$, $l_5 = 400\,\mu m$, and $\theta = 45°$. **g** Numerical simulation of the amplitude of the force on the 2D unit cell for sources 1 and 2 and as a function of frequency. **h** The experimental setup is described in the "Methods" section. **i** The displacements in the $x$- and $y$-directions of the free end (point $A$) of the tilted cantilever beam are plotted for different frequencies and loading directions.

centered main beam acting as the main mass, of two (or more) symmetric tilted cantilevers acting as both converters and subsystems, of a curved beam providing the main restoring spring, and of U-shaped boundaries at the sides and bottom ensuring connectivity between adjacent cells as well as clipping the motion of subsystems. The geometry parameters of the unit cell are defined in Fig. 2c. The length $l$, the thickness of the fixed end $t_1$, the diameter of the free end of the tilted cantilever $2r$, the angle between the main beam and tilted cantilever $\theta$, and the distance between the free end of the tilted cantilever and the fixed boundary $d$ are the key geometry parameters. Whereas lengths $t_1$, $l$, and $2r$ influence the resonant frequencies of the tilted cantilever, angle $\theta$ determines the transformation ratio of subsystem displacements in the direction perpendicular to those along the direction parallel to the external source. The gap distance $d$ has a strong influence on the amplitude of the external stimulus that is required for the

free end of the tilted cantilever to contact with the fixed boundary. Furthermore, the proposed 2D unit cell can be used to compose 2D non-reciprocal and non-Newtonian metamaterials, as illustrated in the inset of Fig. 2c. Of course, the design of three-dimensional unit cells and metamaterials based on the same principle can be conducted as well, as shown in Fig. 2d. In this work, 2D unit cells and corresponding metamaterials are employed to demonstrate non-reciprocal and non-Newtonian mechanical behavior.

We first examine specifically the operation of the converter. Micro-scale unit cells and metamaterials comprising several unit cells were fabricated by two-photon lithography 3D Direct Laser Writing technology with IP-S resin, as illustrated in Fig. 2e, f (see the "Methods" section). As a note, it is difficult to obtain the force versus displacement response of micro samples as a function of frequency directly. Hence, a numerical method was first employed to estimate the

variation of the amplitude of the force for a 2D unit cell (see the "Methods" section). Here, a harmonic excitation with 2 μm amplitude, for both source 1 and source 2, is applied to the structure and a frequency sweep from 10 to 30 kHz is considered, as reported in Fig. 2g. Numerical results indicate that the amplitude of the force is enhanced dramatically when the frequency of source 1 is tuned to the resonant frequency of the tilted cantilevers. As discussed above, this enhancement depends on two factors, resonance of the subsystems and the presence of the fixed boundaries limiting the vibration of the cantilevers. In contrast, the frequency response is constant with frequency and non-resonant if the excitation is applied by source 2. The non-reciprocal numerical results reported in Fig. 2g are in agreement with the discussion in Fig. 1b.

An experimental setup to measure the response of the cantilevers to different excitation frequencies was prepared, as illustrated in Fig. 2h. Orienting samples downward (label 1) or upward (label 2) they can be excited under the conditions described in Fig. 1b. We use a piezoelectric patch as the vibration source and perform the experiment in the dark, the only source of light coming from the light emitting diodes (LED). A signal generator is used to drive the piezoelectric transducer and a pulsed signal is sent to the LED at a slightly higher frequency (details are given in the Methods section). The effective vibration cycles are thus observed with a microscale stroboscopic-like illumination system. The sample considered is shown in the inset of Fig. 2e (see Section 1 in Supplementary Materials). Two different excitation frequencies are applied to the 2D unit cell, 10 kHz (non-resonant frequency) and 23.5 kHz (resonant frequency of the tilted cantilevers). The image-tracking method is used to extract the trajectory in the $x$- and the $y$-direction of point $A$ located at the free end of the tilted cantilever, as shown in Fig. 2e. Results are reported in Fig. 2i. When the external stimulus is applied from time-harmonic source 1 at a non-resonant frequency (10 kHz), point $A$ moves periodically in the $y$-direction, but the displacement in the $x$-direction remains small, which indicates that the converters follow the translation of the source (rigid motion) inducing little elastic vibrations in the cantilever. In contrast, if the frequency of source 1 is set to the resonant frequency (23.5 kHz) of the cantilever, point $A$ oscillates along both the $x$- and the $y$-directions. The displacements of point A along the $x$- and the $y$-direction, relative to the hinge between the main beam and the cantilever, should be equal, for angle $\theta = 45°$. The experimental results for the $y$-direction include the vertical motion of the hinge so that the displacement in the $x$-direction is smaller than that in the $y$-direction. As a whole, the converter transfers the motion of the external source to the motion of the subsystems in a different direction; the induced motion depends strongly on the frequency of the external source.

We now consider the response to time-harmonic source 2 and the same pair of frequencies. We argued before that no displacement of the cantilever should be induced in this case, regardless of the excitation frequency. However, the experimental results in Fig. 2i indicate that source 2 induces frequency-dependent elastic vibrations of the cantilever, as source 1 does, though with a smaller amplitude. The primary explanation for this deviation from theory is that the main beam is an elastic body in reality, not a rigid body. As a result, elastic deformations of the main beam can induce elastic vibrations of the cantilever. In Section 2 of the Supplementary Material, we verify via numerical simulations that the elastic vibrations of the cantilever for source 2 weaken as the stiffness of the main beam increases. Supplementary Video 1 shows an experimental movie of the deformation of the metamaterial under an external stimulus; for different frequencies and directions of excitation. Globally, it is clear that the mechanical response depends on the direction of the applied stimulus, a signature of non-reciprocity.

## Non-reciprocal and non-Newtonian mechanical responses of the metamaterials

In the previous sections, we designed subsystems that are able to reach the fixed boundaries depending on the frequency and that can be excited in a non-reciprocal way. The ultimate goal of this work, however, is to obtain a non-Newtonian, or velocity-dependent, mechanical metamaterial response. For this purpose, the boundaries should be able of fixing the subsystems after they come into contact with them, as a result of a transient excitation. The shape of the fixed boundary is chosen so that it is able to fix the subsystem during the loading process and release it during the unloading process, as shown in the enlarged illustration of Fig. 3a. It is worth noting that this type of boundary is a self-locking system, that is, the constraints in the subsystem increase with the deformation of the entire/sub-system.

Three models (models i–iii in Fig. 3a) with different parameters and configurations are proposed in this section. Their velocity and loading-direction-dependent mechanical responses are studied numerically and experimentally in the following. For model i, the reaction force (in Fig. 3a) increases linearly during the whole low-speed (0.002 m/s) compression loading process, as only the curved beam (valid spring) is deformed; that is, all subsystems of model i are without any elastic deformation in this case. In contrast, subjected to high-speed compression loading (2 m/s), the mechanical response changes remarkably, as indicated by the green curve in Fig. 3a. In this case, the entire compression process can be divided into three steps, from A to C. In step A, the subsystems start to move toward the boundaries but without contacting them yet, so that the mechanical response of the overall system looks identical to the stationary compression situation. Because of inertia, however, a transient elastic wave propagates along the cantilever, causing it to expand in the transverse direction. In step B, the cantilevers touch the boundaries and are fixed by them. Subsequently, a compression-dominated elastic deformation occurs in the subsystem, causing a large change in the reaction force and thus in the effective stiffness that raises dramatically. The compression-deformation process then reaches step C. At this step, bending will proceed in the thin part (the fixed end or hinge) of the cantilever so that the reaction force decreases at the beginning of step C. Afterward, the bottom side of the thin part of the cantilever fully contacts the central main beam and hence improves the effective properties of the cantilever, causing the reaction force to increase again. In order to validate those numerical results, the corresponding impact experiments were carried out by using 3D-printed centimeter-size TPU samples (details are available in the Methods section), as illustrated in Fig. 3b. It can be seen that the experimental results are in good agreement with simulation results, i.e., the tilted cantilevers undergo rigid-body translation motion merely in the low-speed loading case but are deformed and fixed by the boundaries under high-loading speed circumstances. It is worth noting that the force versus loading displacement curves of Fig. 3 are numerical results; the corresponding experimental results are presented in section 3 of the supplementary materials.

Significantly, it is clear that the mechanical response of model i depends highly on the loading speed of the external stimulus, i.e., the signature of non-Newtonian elasticity. Compared to low-speed loading, the reaction force for model i increases by more than an order of magnitude (from 22 to 550 N) under high-speed loading. We can further improve the non-Newtonian response of the metamaterial by increasing the number of subsystems, i.e. of tilted cantilevers, as shown with model ii. The corresponding reaction force doubles in Fig. 3a with twice more cantilevers and the same stimulus. Flipping the sign of angle $\theta$ and inverting the direction of the external source, we also can make the metamaterial stretchable, as shown for model iii in Fig. 3a.

We now consider excitation by source 2 connected to the main restoring spring. An external stimulus with variable loading speed is

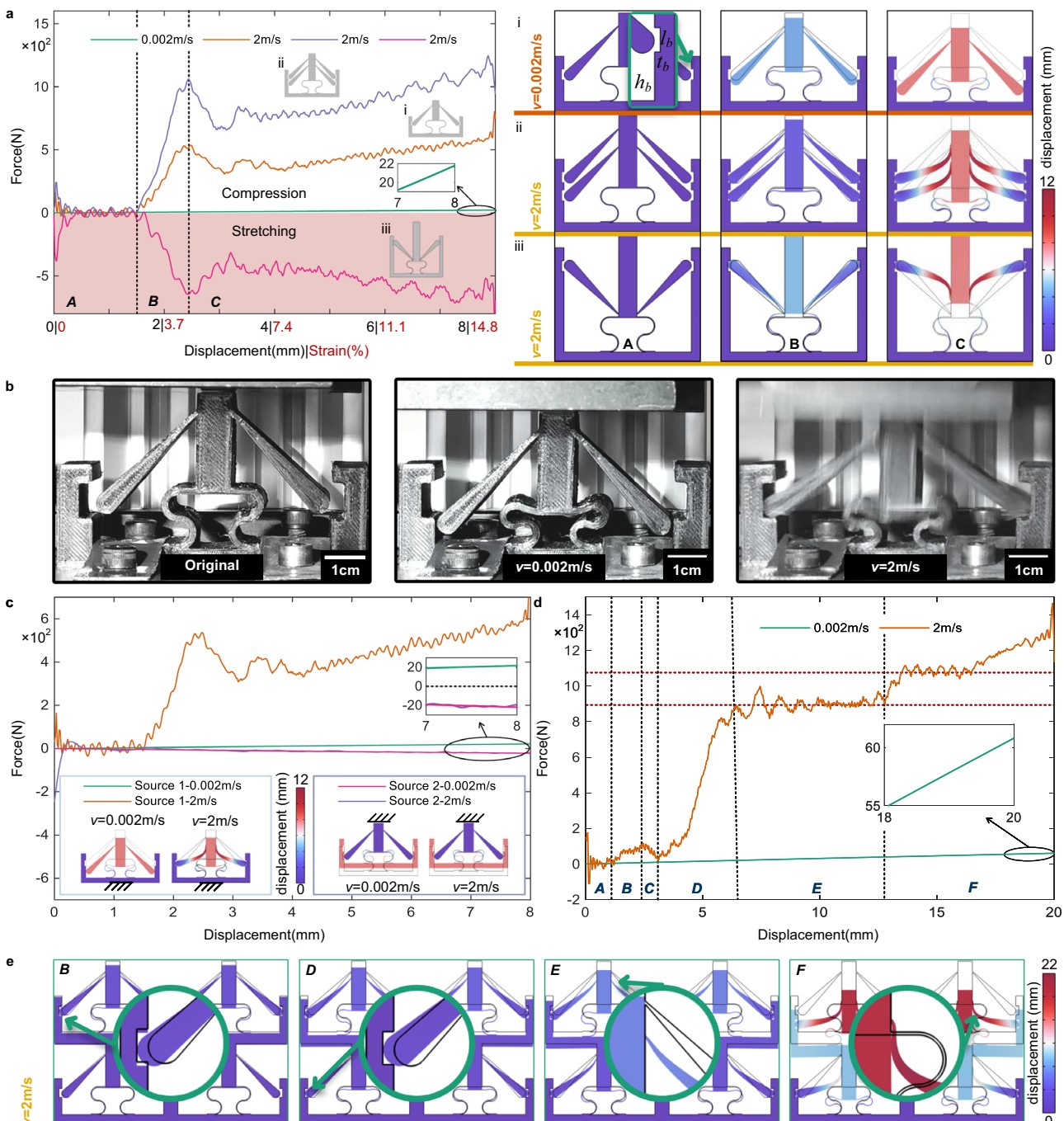

**Fig. 3 | Non-reciprocal and non-Newtonian mechanical response of the metamaterials for different impact velocities. a** The velocity-dependent mechanical response for different unit cells is shown as a function of the loading displacement (also depicted as the strain) for source 1. For the three models (i–iii), the geometrical parameters are $l = 38.2$ mm, $l_2 = 7$ mm, $l_3 = t_3 = t_4 = r_1 = 5$ mm, $l_5 = 80.2$ mm, $l_b = 6$ mm, $t_1 = 0.57$ mm, $t_2 = 0.5$ mm, $t_5 = 5$ mm, $t_b = 1.4$ mm, $r = 3$ mm, $d = 0.1$ mm, and $h_b = 12$ mm. For model i and ii, $l_1 = 22.5$ mm and $\theta = 45°$. For model iii, $l_1 = 1$ mm and $\theta = 135°$. $v$: the velocity of loading. **b** Experimental tests of the mechanical response are performed at different velocities for macroscopic samples. We use the following geometrical parameters: $t_5 = 10$ mm, $h_b = 9$ mm, $t_b = 4$ mm, $l_b = 11$ mm, $d = 1$ mm. Other parameters are the same as for model i. **c** The non-reciprocal and non-Newtonian mechanical response of the unit cell model i. **d** The velocity-dependent mechanical response of the metamaterial composed of 2 × 2 model i unit cells is shown. Different regions of interest are labeled from A to F. **e** Snapshots are presented for the deformation process under a 2 m/s impact test at different important moments (steps B, D–F).

applied to model i and the displacement is plotted versus the force in Fig. 3c. The responses for low-speed (0.002 m/s) and high-speed (2 m/s) compression loading are similar, except for some fluctuations at the beginning of the loading in the high-speed case. These fluctuations result from the impact resistance that also can be observed at the beginning of step *A* for source 1 in Fig. 3a. Significantly, whatever the loading speed, the main spring deforms but not the tilted cantilevers.

Hence, the response of the structure is non-reciprocal in addition to being non-Newtonian.

Furthermore, metamaterials can be composed of the elementary unit cell. The mechanical response of a metamaterial composed of 2 × 2 model i unit cells is presented in Fig. 3d. The response of the metamaterial is inherited from the response of the unit cell but avoids the softening of the latter. The dynamic response in Fig. 3d can be

divided into 6 steps, labeled from A to F. In step A, the curved beam in the top layer deforms similarly as in step A for the unit cell model i. Afterward, in step B the tilted cantilevers in the top layer contact and are fixed by the boundaries, so that the reaction force increases sharply. It is worth noting that at this step the effective stiffness varies due to the cantilevers in the bottom layer still being separate from the fixed boundaries. As a result, the force transmits rapidly from the top layer to the bottom layer in step C. For this reason, the cantilevers in the bottom layer start to move toward the boundaries and the reaction response of the entire system slightly diminishes. Then the force and the effective stiffness rise dramatically in step D thanks to all cantilevers being fixed by the boundaries. The deformation process is similar to step B for the model i unit cell, i.e., the deformation characteristic of the cantilevers is compression-dominant. Next, because of the bending of the thin part (the fixed end) of the cantilevers, the reaction force fluctuates slightly in step E. Finally, the lower surface of the thin part of the cantilever fully contacts with the central main beam. The effective length of the cantilever reduces and the reaction force further increases (see step F in Fig. 3e), i.e., the reaction force raises step-by-step under dynamic loading, as shown by the red dotted lines in Fig. 3d. Animations of the deformation process are available in the Supplementary Video.

## Potential applications of non-Newtonian metamaterials to aerospace

Let us discuss potential situations where a non-Newtonian solid metamaterial could be of help. In the aerospace industry, satellite docking is a delicate process because of the strict requirements on space stations and of the limitations of the carrying capacity of rockets. In principle, the relative velocity between two satellites should be small during the docking process. Nevertheless, in certain emergency conditions, e.g., if one of the satellites is uncontrollable and has a high

relative velocity, an impact could cause damage to both satellites. The proposed stretchable and compressible non-reciprocal and non-Newtonian mechanical metamaterials could find application as a connector between both satellites during the docking process, as depicted in Fig. 4.

In the normal case, the influence of the metamaterials on the docking process can be ignored due to the small relative velocity (0.3 m/s). The metamaterials just show an extra soft mechanical behavior, as demonstrated in Fig. 4a. In contrast, if the relative velocity exceeds a critical value, the two satellites will not be able to connect with each other before the relative velocity drops to 0 m/s since the connector (metamaterial) will become stiff and will become able to absorb the excess kinetic energy, as shown in Fig. 4b. It is clear that the energy absorption capacity of the metamaterial under low-speed loading depends on the elastic deformation of the curved beams, but under and high-speed impact it depends on the response of the tilted cantilevers. Simulation details are given in Section 4 of the Supplementary Material and an animation of the docking process is provided in the Supplementary Video (in Section 3 as well).

Similarly, non-Newtonian solid metamaterials also have potential applications in the transportation industry, for instance in speed bumps. If the speed of the car is smaller than the required critical velocity, the influence of the speed bump can be ignored and will not introduce any discomfort to the driver and the passengers, thanks to the stationary mechanical properties being weak. In contrast, the speed bump will become stiff when the speed of the car exceeds the required velocity. All in all, there are core advantages of the proposed metamaterials that need to be emphasized. First of all, both static and dynamic mechanical properties of the metamaterials are highly customizable. For example, it is easy to obtain different mechanical behaviors by modifying the valid spring (the curved beam) $K$, the number of tilted cantilevers, and the thickness of the fixed end of the

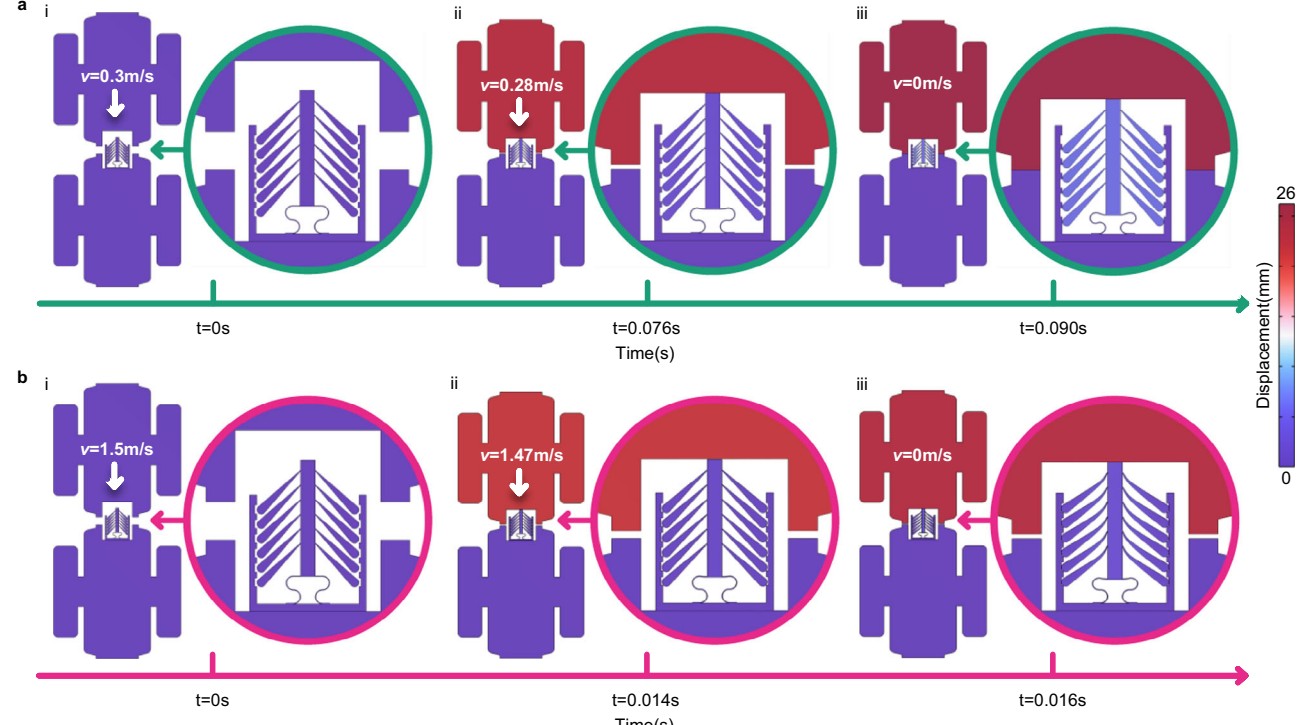

**Fig. 4 | Potential applications of the non-reciprocal and non-Newtonian mechanical metamaterials for satellite docking. a** In a normal situation, the relative velocity between two satellites is quite small (e.g., 0.3 m/s) and the influence of the metamaterials on docking can be ignored as they are soft. **b** Under an emergency circumstance where the relative velocity between the two satellites is larger than the critical value, the metamaterials become stiff and are able to absorb the excess kinetic energy to avoid shock and damage to both satellites.

cantilevers so as to change the critical loading speed between the static and dynamical response. Second, the metamaterials are smart materials for the loading speed, implying that additional velocity sensors are unnecessary and that the complexity of the entire system can be reduced. They could also reduce the weight and improve the reliability of the system. Finally, non-Newtonian solid metamaterials can play a protective role in the packing of delicate objects, since the step-by-step increase of the reaction force under dynamic loading can effectively reduce the peak stress.

## Discussion

Summarizing, we have demonstrated how stretchable and compressible non-reciprocal and non-Newtonian elastic/mechanical metamaterials can be realized by combining the concept of local resonances and of fixing boundaries. Our model design consists of a main system with an effective mass and a valid spring, of subsystems (also described with an effective mass-spring model) connected to the main system through converters, and of fixed boundaries. An off-axis loading tilted cantilever with a concentrated mass at the free end was proposed as an example of a converter. 2D and 3D mechanical metamaterials were presented and studied numerically and experimentally. Experimental results show that the converter is able to transfer the movement of the main system to the subsystems and to change its direction, this transfer is frequency dependent as well. Computational models and impact experiments for both unit cells and metamaterials demonstrate the stretchable and compressible non-reciprocal and non-Newtonian mechanical response. The stiffness effectively changes by more than an order of magnitude as a function of the loading velocity. Non-Newtonian solid metamaterials are expected to find potential engineering applications in miscellaneous fields, involving for instance speed bumps, satellite docking, and soft robotics.

## Methods

### Fabrication

The samples at the micro-scale were fabricated by a commercial 3D printer (Photonic Professional GT+, Nanoscribe GmbH) that is based on two-photon lithography 3D Direct Laser Writing. The customized commercial negative tone IP-S resin (Nanoscribe GmbH) was employed as the raw material. The Young's modulus, Poisson's ratio, and density of this type of resin are $E = 4$ GPa, $\nu = 0.43$, and $\rho = 1.00$ g/cm$^3$, respectively. Moreover, we set the slicing distance to 1 μm and the hatching distance to 0.5 μm. A drop of IP-S resin was deposited on an ITO-coated sodalime glass substrate with dimensions 25 mm × 25 mm × 0.7 mm and photopolymerized with a femtosecond laser operating at $\lambda = 780$ nm and a ×25-objective. For the 2D micro-scale unit cell samples, the main body and the fixed boundaries (except for two tilted cantilevers) were printed from the bottom to the top first, then the tilted cantilevers were fabricated from the top to the bottom (i.e., from the fixed end to the free end). The 3D micro-scale composite samples were printed unit cell by unit cell based on the method described above. After printing, the samples were developed for 30 min in propylene glycol methyl ether acetate (PGMEA) solution to remove the unexposed photoresist and rinsed for 5 min in Isopropyl alcohol (IPA) to clear the developer. A laser power of 100 mW and a galvanometric scan speed of 100 mm/s were used for the whole fabrication process.

The samples at the centimeter scale were printed by a commercial 3D printer (Ultimaker 3, Ultimaker BV) that is based on the fused deposition modeling (FDM) principle. The raw material is black TPU 95A (thermoplastic polyurethane, Ultimaker BV) with Young's modulus $E = 26$ MPa and density $\rho = 1.22$ g/cm$^3$. To get extra fine printing quality, the layer height, wall thickness, top/bottom thickness, printing temperature, printing speed, and infill density were set as 0.06, 1, 1 mm, 228 °C, 20 mm/s, and 100%, respectively.

### Test

For frequency-domain tests, based on the principle of stroboscopy, the experimental setup shown in Fig. 2g was built and used. This type of setup consists of a frame, a piezoelectric patch, a signal generator, a 50 times amplifier, a LED light, a 50 times objective, and a computer. The way they are connected to each other is demonstrated in Fig. 2g. The entire setup works in a dark environment and the only source of light is a LED. The piezoelectric transducer works as an external source fed by a harmonic signal with 10 kHz/23.5 kHz frequencies under 4 V which are generated by a signal generator and amplified by a 50 times amplifier. Moreover, the LED was modulated by a 20% pulse signal with 10.0004 kHz/23.5004 kHz frequencies under 4 V. This way, the 50 times objective is able to capture the periodic motion of the free end of the tilted cantilevers. this periodic motion is recorded by the computer as a video with a resolution of 2560 × 1920 and 4.4 frames per second. Finally, MATLAB 2018b was employed to process images and extract the trajectory of the free end of the cantilevers.

For the high-speed (impact) tests, a free-fall impactor in which all degrees of freedom have been constrained except those in the vertical direction was employed. Based on the relation between the velocity and the height of the free-fall, the height of the free-fall impactor was set to 20 cm to obtain a speed of 2 m/s. At the same time, a camera parallel to the sample is employed to record the impact process.

### Simulations

The Solid Mechanics module in the commercial finite element method software COMSOL 6.0 was employed for numerical simulations. Identical 3D geometry models were built for simulations and for 3D printing to reduce the difference between experiments and simulations. First, we used contact pairs with a penalty formulation to define the contact between the free end of cantilevers and the fixed boundaries. After that, for the simulation shown in Fig. 2g, a harmonic excitation is applied to the structure as either source 1 or 2. To obtain convergent results for the force amplitude, the response of 30 cycles (with each cycle decomposed in 100 steps) is calculated for each frequency. For the simulations shown in Fig. 3, a prescribed velocity with constant loading speed along the $y$-direction was applied at the top (or at the bottom) of the unit cell (or metamaterial), and all degrees of freedom of the bottom (or top) boundary outer surface were fixed at the same time. A free tetrahedral mesh with a predefined extra fine element size was used for the entire structure. Then, we utilized the time-dependent module in the study section to finish the calculation process. The total simulation time is governed by the ratio of loading displacement to velocity. The entire process includes 500 steps and a physics-controlled tolerance was used for simulations. Finally, in the "Results" section, the surface integration sub-option under the derived values was employed to output the reaction force in the $y$-direction. It is worth noting that the SMOOTH function in MATLAB 2018b was used to process the original simulation results to reduce fluctuations. The influence of gravity is neglected in all simulations.

## Data availability

All the data supporting the conclusions of this study are included in the article and the Supplementary Information file. Source data are provided with this paper (https://doi.org/10.6084/m9.figshare.23389370). Source data are provided with this paper.

## Code availability

The computer code that supports the findings of this study is available from the corresponding author on request.

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

## Acknowledgements

This work was supported by the National Key Research and Development Program of China (grant number 2022YFB3707800, B.W.), the EIPHI Graduate School of UBFC [grant number ANR-17-EURE-0002, V.L.], the French Investissements d'Avenir program, in part by the ANR PNanoBot (ANR-21-CE33-0015, M.K.) and ANR OPTOBOTS project (ANR-21-CE33-0003, M.K.), the french RENATECH network and its FEMTO-ST technological facility (V.L.), the National Natural Science Foundation of China [grant number 11972008, B.W.], and the China Scholarship Council (grant number 202106120088, L.W.). In addition, we would like to especially thank Dr. Lin Sun (Harbin Institute of Technology) for preparing the samples for the impact experiments.

## Author contributions

L.W. conceived the study and carried out the model design, experimental testing, and computational analysis, supervised by B.W., V.L., and M.K., G.U. prepared samples with 3D printing. J.A.I.M. built the experimental setup. All authors worked together to write and revise the manuscript.

## Competing interests

The authors declare no competing interests.
