## [Peer Review File · Nature Communications]

Non-reciprocal and non-newtonian mechanical metamaterialsREVIEWER COMMENTS

Reviewer #1 (Remarks to the Author):

The authors present a very interesting idea of a material, which is both non-reciprocal, and non-Newtonian in the sense that it has a velocity-dependent mechanical response. The toy model consists of a main mass and spring system, subsystem masses and springs, and converters, which provide a frequency-dependent transformation of the direction of motion between the main system and the subsystems. The authors validate the actual metamaterial model using Finite Elements simulations and experimentally.

The work is innovative and significant to the field, and I do recommend publication. I have a question, though:

Is it possible to formulate the explicit equations of motion for the toy model system? This could improve the clarity of the presentation, as one could directly simulate the underlying dynamics in the various regimes.

Reviewer #2 (Remarks to the Author):

The authors present a theoretical design strategy for structures that are both non-reciprocal (their response depends on the actuation location) and non-Newtonian (their response depends on the actuation speed). The designs consist of a collection of springs and connecting elements that join various motion-constrained masses.

The authors show a potential 2D unit cell design and realize it experimentally using two-photon lithography and 3D-printing on micrometric and centimetric scales, respectively. The authors also present a potential 3D unit cell design, and realize it on the micrometer scale.

The displacement-frequency response of the micrometric 2D structure is studied experimentally. The force-displacement response of several variations of the centimetric 2D structure is studied numerically. Qualitative experimental results match the deformation response of the simulations. The authors propose an application for the unit cell as a macroscopic shock-absorber between two objects, and present numerical results to support this proposal.

I believe the design strategy presented is novel and significant. Specifically, the work investigates structures that combine variable mass density, stiffness, and motion constraints, in order to tune mechanical properties. I am not aware of other work in the field of mechanical metamaterials that uses this particular combination.

In their introduction, the authors provide a brief but thorough review of existing work on reciprocity-breaking and speed-dependent responses in architected materials. I recommend recent work on using viscoelasticity to design speed-dependent mechanics (<https://doi.org/10.1063/5.0094224>, Dykstra et al, APL Mater. 2022), and I encourage the authors to see if this work helps bolster the novelty of their findings.

The data, models, and analysis methods used are presented clearly. I do note that the authors mostly use numerical simulations to support their claims. It is unclear to me to what extent experimental samples can reproduce both the simulated deformations and forces accurately; it would be valuable if the authors could comment on this.

Minor issues:

- color choices in Fig.2g and Fig.2i lead to some confusion. I recommend using distinct color sets for sources 2 and 1 and x/y-directions (they are currently both blue and orange).
- Please indicate point A referred to in Fig.2i (e.g. in Fig.2h).
- While the deformation response of the 3D-printed structure visually matches that of the simulations, no experimental force response is reported. It would be good to point this out clearly in the main text.
- Adding the results for model (i) in Fig.3a to the graph in Fig.3c would help highlight the combination of non-reciprocity and speed-dependence found in the simulations, and give a sense of scale for both effects.

I believe the results presented present a novel and potentially significant strategy for the design of metamaterials with tunable non-reciprocity and speed-dependent behaviour. If the authors can address the comments above, I would be happy to recommend this paper for publication in Nature Communications.

Response to the Reviewer's Letter

Article title: *Non-reciprocal and Non-Newtonian Mechanical Metamaterials*

Journal: *Nature Communications*

Manuscript number: *NCOMMS-23-10939*

In this response letter, we repeat all comments of the two reviewers in black, we respond to them in red, and we highlight changes made to the manuscript and supplementary materials in blue.

Reviewer #1 (Remarks to the Author):

The authors present a very interesting idea of a material, which is both non-reciprocal, and non-Newtonian in the sense that it has a velocity-dependent mechanical response. The toy model consists of a main mass and spring system, subsystem masses and springs, and converters, which provide a frequency-dependent transformation of the direction of motion between the main system and the subsystems. The authors validate the actual metamaterial model using Finite Elements simulations and experimentally.

The work is innovative and significant to the field, and I do recommend publication.

Question: I have a question, though: is it possible to formulate the explicit equations of motion for the toy model system? This could improve the clarity of the presentation, as one could directly simulate the underlying dynamics in the various regimes.

Response:

We thank the reviewer for this suggestion that prompted us to formulate mathematically the toy model. The following text has been added to the main text.

Source 1 is characterized by an applied force on the main mass that is a function of time, $F_1(t)$. For instance, that applied force can be time-harmonic, i.e. of the form $F_1(t) = A \exp(i\omega t)$ with driving angular frequency ω and amplitude A . The center of mass of the main system, considered a rigid body, has displacements (x, y) . By symmetry, we assume $x = 0$ at all times. The displacements of the subsystems are represented by degrees of freedom $(x_i, y_i), i = 1 \dots N$. Let us first consider the case that no

Figure 1. Toy model of the metamaterial.

internal resonances of the sub-systems are excited, i.e., ω remains far from any the resonance frequencies $\omega_i = \sqrt{K_i/M_i} = 2\pi f_i$. Then we can assume $x_i = 0$ and $y_i = y$. The motion of the main mass satisfies the dynamical equation

$$\left(M + \sum_{i=1}^N M_i \right) \ddot{y} + Ky = F_1(t), \quad (1)$$

Damping is ignored in the whole system and the masses of all springs are neglected for simplicity. Equation (1) can easily be solved as a function of time.

Next, we include the motion of the sub-systems, i.e. frequency ω can approach one or several of the resonance frequencies ω_i . Each of the sub-systems satisfies a dynamical equation

$$M_i \ddot{x}_i + K_i x_i = F_i^r(t) = \gamma_i(\omega) A \exp(i\omega t). \quad (2)$$

$F_i^r(t)$ is the x -component of the reaction force introduced by the resonance of the subsystem. $\gamma_i(\omega)$ is a conversion factor of source 1 to the motion of sub-system i , expressing the assumed linearity of the reaction force with the applied force. As a note $\gamma_i(\omega)$ may not be precisely known if the properties of the converter are not defined. The reaction forces act on the main rigid mass, but we also have to include in the model the forces resulting from contact of the sub-systems with the fixed external boundaries, noted $F_i^b(t)$. The $F_i^b(t)$ are again functions of the applied force, e.g. depend on A and ω , but this time in a non-linear way. The motion of the main mass now satisfies

$$\left(M + \sum_{i=1}^N M_i \right) \ddot{y} + Ky = F_1(t) + \sum_{i=1}^N \alpha_i F_i^r(t) + \sum_{i=1}^N F_i^b(t), \quad (3)$$

Factors α_i are introduced to express the linear relationship of the y -component of the reaction forces with their x -component.

All in all, equations (2) and (3) provide $N + 1$ equations for the $N + 1$ unknowns (y, x_i) and the model can be solved. In practice, however, it would be necessary to provide accurate values for parameters $\gamma_i(\omega)$ and α_i , and an expression for forces $F_i^b(t)$. Hence, the toy model is used here only to discuss the origin of

the dynamical non-Newtonian properties, but we use numerical simulations based on continuum linear mechanics in the following.

Source 2 can also be characterized by an applied force, but on the main spring and not on the main mass, that is a function of time, $F_2(t)$. Denoting ξ the elongation of the main spring, Hooke's law yields

$$K\xi = F_2(t). \quad (4)$$

Applied force $F_2(t)$ is transferred to the main mass but the latter does not move, because of the assumption of rigidity and of the application of a clamping boundary condition at the top, hence $y = 0$ and $y_i = 0$. This is of course a strong assumption requiring a very soft main spring and a very rigid main mass, but a direct expression of the mechanical non-reciprocity of the system.

Reviewer #2 (Remarks to the Author):

The authors present a theoretical design strategy for structures that are both non-reciprocal (their response depends on the actuation location) and non-Newtonian (their response depends on the actuation speed). The designs consist of a collection of springs and connecting elements that join various motion-constrained masses. The authors show a potential 2D unit cell design and realize it experimentally using two-photon lithography and 3D-printing on micrometric and centimetric scales, respectively. The authors also present a potential 3D unit cell design, and realize it on the micrometer scale.

The displacement-frequency response of the micrometric 2D structure is studied experimentally. The force-displacement response of several variations of the centimetric 2D structure is studied numerically. Qualitative experimental results match the deformation response of the simulations. The authors propose an application for the unit cell as a macroscopic shock-absorber between two objects, and present numerical results to support this proposal.

I believe the design strategy presented is novel and significant. Specifically, the work investigates structures that combine variable mass density, stiffness, and motion constraints, in order to tune mechanical properties. I am not aware of other work in the field of mechanical metamaterials that uses this particular combination.

Question: In their introduction, the authors provide a brief but thorough review of existing work on reciprocity-breaking and speed-dependent responses in architected materials. I recommend recent work on using viscoelasticity to design speed-dependent mechanics (<https://doi.org/10.1063/5.0094224>, Dykstra et al, APL Mater. 2022), and I encourage the authors to see if this work helps bolster the novelty of their findings.

Response:

Thank you for recommending this excellent work. We have introduced it in the main text as "The interaction between mechanical instabilities and viscoelasticity was discussed and utilized by Dykstra *et al.*⁵⁴ to obtain loading rate-dependent functionalities."

Question: The data, models, and analysis methods used are presented clearly. I do note that the authors mostly use numerical simulations to support their claims. It is unclear to me to what extent experimental samples can reproduce both the simulated deformations and forces accurately; it would be valuable if the authors could comment on this.

Response:

In order to answer the reviewer's prompt to support more firmly our claims with experimental results, we have added the following results and discussions in the Supplementary Material.

To evaluate the non-Newtonian mechanical performance of the considered metamaterials, the following quasi-static compression and impact test of the metamaterials were performed, as shown in Fig. S3. Three types of samples were designed and fabricated, named sample A-C and illustrated in Fig. S3d-*i*, e-*i*, and f-*i*, respectively. The geometrical parameters for sample A are: $l = 38.2$ mm, $l_2 = 4.3$ mm, $t_3 = 14$ mm, $l_3 = t_4 = r_1 = 5$ mm, $l_5 = 98.8$ mm, $l_b = 10$ mm, $t_1 = 1.5$ mm, $t_2 = 1.5$ mm, $t_5 = 16$ mm, $t_b = 4$ mm, $r = 3$ mm, $d = 1$ mm, $h_b = 9$ mm, $l_1 = 22.5$ mm, and $\theta = 45^\circ$. Sample B is identical to sample A but without the fixed boundaries. Sample C is identical to sample B but without the tilted cantilevers (i.e., the subsystems).

First, the INSTRON 3344 mechanical measurement machine was employed for the quasi-static compression test of sample A. The static compression deformation and corresponding force versus compression displacement curve are shown in Fig. S3a and b. It is clear from Fig. S3a that the subsystems (i.e., the tilted cantilevers) are not contacting with the fixed boundary during the whole compression process. In other words, in this case, the static mechanical properties of the metamaterial only depend on the S-shaped beams (corresponding to the main spring K of the toy model). The maximum reaction force value is 5.5 N, as demonstrated in Fig. S3b.

In order to assess the dynamic behavior of the metamaterials, we built the experimental setup for the impact test shown in Fig. S3c. The setup consists of an acceleration sensor, an impactor, a signal processor, and a fixed frame. The sampling frequency of the acceleration sensor is 2 KHz. The mass of the impactor is 4.6 kg and it can free-fall (the height is set to 60 cm) along the vertical frames. The acceleration data of the impactor is generated by the signal processor directly. The acceleration of the impactor is used to obtain the reaction force (multiplying it by the mass of the impactor) of the metamaterials.

The impact response for sample A is illustrated in Fig. S3d. Force as a function of time includes 4 peaks, as shown in Fig. S3d-*ii*. As a remark, the last 3 peaks are caused by bounces of the impactor. It is also worth noting that the first peak corresponds to the collapse of the metamaterial, at which time the main central beam (corresponding to the main mass M in the toy model) contacts the bottom of the metamaterial, as demonstrated in the insert of Fig. S3d-*ii*. Before the collapse, the tilted cantilevers are deformed (see Fig. S3d-*iii*) and the mechanical response is shown in Fig. S3d-*iii*. There also a peak that can be observed when the force reaches 570 N, after which the force decreases due to the nonlinear mechanical response of the cantilevers under large deformations. The simulation results show a very similar trend in the nonlinear mechanical response, as presented in Fig. 3 of the main text. Eventually, the force increases dramatically because of the occurrence of the collapse of the whole structure. Significantly, and as compared to the static situation, the mechanical response of the metamaterial approximately increases 100 times (from 5.5 N to 570 N) in the dynamic case, which is the signature of the non-Newtonian behavior.

To investigate the cause of the formation of the non-Newtonian properties of the metamaterial, impact tests for samples B and C were carried out experimentally, and consistent results were obtained and are shown in Fig. S3e and f, respectively. It is clear that both samples B and C have impact responses similar to sample A. However, the peak force (before collapse) of these samples is different: 404 N for sample B (see Fig. S3e-*iii*) and 357 N for sample C (see Fig. S3f-*iii*). The difference in the peak forces for samples A and B (570 N – 404 N = 166 N) is an indication of the reaction of the fixed boundaries. Similarly, the difference in the peak forces for samples B and C (404 N – 357 N = 47 N) expresses the response of the subsystems (the tilted cantilevers). Furthermore, the unequal peak forces for samples A and C (570 N – 357 N = 213 N) proves the influence of the subsystems and the fixed boundaries. As a whole, these observations are consistent with the outcome of the toy model, i.e. that the non-Newtonian behavior is introduced by two factors, one being the resonances of the subsystems and the other being the influence of

the fixed boundaries.

Figure S3. Experimental validation of the non-Newtonian mechanical behaviors of the considered metamaterial. a Quasi-static deformation of the metamaterial (sample A, the compression displacement of this moment is 5 mm). **b** The corresponding force versus displacement curve of sample A. **c** Experimental setup for the impact tests. **d-f** The photos of sample A-C and corresponding impact responses.

For completeness, it should be realized there are some differences between the simulation results of

Fig. 3 in the main text and the experimental results of this section. It is not easy in practice to closely reproduce experimentally the numerical results because of the following issues. First, the elastoplasticity of the raw material was ignored in the simulations, since we wanted to highlight the operation of the structure. However, elastoplasticity influences the experimental mechanical response, especially for large compressive deformations. Second, the fracture of the parent material is also not taken into consideration in the numerical model. In fact, we initially tried to use PLA as the raw material to print the samples. Nevertheless, the tilted cantilevers broke during the experimental impact tests. Third, in the experiment, we need a larger rectangular opening hole on the fixed boundary to fix the free end of the cantilever beam, because larger rectangular opening holes provide greater tolerance for experimental error. For example, due to experimental errors (off-axis impact) or manufacturing errors (mass or density of the cantilever beams on both sides are not exactly the same), the impact test may cause a slight asymmetric response of the left and right cantilever beams. The larger rectangular opening holes extend the length of step-A in the displacement-force curves in Fig. 3a of the main text. Fourth, in numerical simulations, the impact velocity was kept constant during the impact process. However, in experiments, the velocity of the impactor actually decreases with time during the impact process.

Minor issues:

Question: - color choices in Fig.2g and Fig.2i lead to some confusion. I recommend using distinct color sets for sources 2 and 1 and x/y-directions (they are currently both blue and orange).

Response:

We have now used distinct colors for different curves in Fig. 2i of the main text.

Figure 2. h The experimental setup is described in the Methods section. i The displacements in the x- and y-directions of the free end (point A) of the tilted cantilever beam are plotted for different frequencies and loading directions.

Question: - Please indicate point A referred to in Fig.2i (e.g. in Fig.2h).

Response:

We now indicate the location of the reference point A in Fig. 2h in the main text.

Question: - While the deformation response of the 3D-printed structure visually matches that of the simulations, no experimental force response is reported. It would be good to point this out clearly in the main text.

Response:

We now point this fact out in the revised main text by adding the statement "It is worth noting that the

force versus loading displacement curves of Fig. 3 are numerical results; the corresponding experimental results are presented in section 3 of the supplementary materials".

Question: - Adding the results for model (i) in Fig.3a to the graph in Fig.3c would help highlight the combination of non-reciprocity and speed-dependence found in the simulations, and give a sense of scale for both effects.

Response:

Figure 3. c The Non-reciprocal and non-Newtonian mechanical response of the unit cell model *i*.

We have now changed Fig. 3c in the main text.

I believe the results presented present a novel and potentially significant strategy for the design of metamaterials with tunable non-reciprocity and speed-dependent behaviour. If the authors can address the comments above, I would be happy to recommend this paper for publication in Nature Communications.

REVIEWERS' COMMENTS

Reviewer #1 (Remarks to the Author):

The authors answered my question properly and I therefore recommend publication.

Reviewer #2 (Remarks to the Author):

I thank the authors for their thorough response and revisions. The authors have addressed all issues and questions pointed out in my previous report, and I am pleased to recommend the work for publication in Nature Communications.